Predicting epidermal growth factor receptor (EGFR) mutation status in non-small cell lung cancer (NSCLC) patients through logistic regression: a model incorporating clinical characteristics, computed tomography (CT) imaging features, and tumor marker levels

Hao Jimin 1
Liu Man 2 3
Zhou Zhigang 4
Zhao Chunling 1
Dai Liping 2 lpdai@hotmail.com
Ouyang Songyun 1 ouyangsy@163.com
1 Department of Respiratory and Sleep Medicine, The First Affiliated Hospital, Zhengzhou University , Zhengzhou, Henan , China
2 Henan Institute of Medical and Pharmaceutical Sciences & Henan Key Medical Laboratory of Tumor Molecular Biomarkers, Zhengzhou University , Zhengzhou, Henan , China
3 Laboratory of Molecular Biology, Henan Luoyang Orthopedic Hospital (Henan Provincial Orthopedic Hospital) , Zhengzhou, Henan , China
4 Department of Radiology, The First Affiliated Hospital, Zhengzhou University , Zhengzhou, Henan , China
Zhang Xin
Electronic publication date: 2024 Dec 3
Publication date: 2024
Volume: 12
Electronic Location ID: e18618
Received 2024 Jul 16; Accepted 2024 Nov 9
Copyright: © 2024 Hao et al.
Copyright year: 2024
Copyright holder: Hao et al.
License: This is an open access article distributed under the terms of the Creative Commons Attribution License, which permits unrestricted use, distribution, reproduction and adaptation in any medium and for any purpose provided that it is properly attributed. For attribution, the original author(s), title, publication source (PeerJ) and either DOI or URL of the article must be cited.
License URL: https://creativecommons.org/licenses/by/4.0/

Keywords: Non-small cell lung cancer (NSCLC), Epidermal growth factor receptor (EGFR), Clinical characteristics, CT imaging features, Tumor marker levels, Logistic regression

Funding: National Natural Science Foundation of China U1804195, 81800091 This research was funded by the National Natural Science Foundation of China (No. U1804195) and the National Natural Science Foundation of China (No. 81800091). The funders had no role in study design, data collection and analysis, decision to publish, or preparation of the manuscript.

==============================
Background

Approximately 60% of Asian populations with non-small cell lung cancer (NSCLC) harbor epidermal growth factor receptor (EGFR) gene mutations, marking it as a pivotal target for genotype-directed therapies. Currently, determining EGFR mutation status relies on DNA sequencing of histological or cytological specimens. This study presents a predictive model integrating clinical parameters, computed tomography (CT) characteristics, and serum tumor markers to forecast EGFR mutation status in NSCLC patients.

Methods

Retrospective data collection was conducted on NSCLC patients diagnosed between January 2018 and June 2019 at the First Affiliated Hospital of Zhengzhou University, with available molecular pathology results. Clinical information, CT imaging features, and serum tumor marker levels were compiled. Four distinct models were employed in constructing the diagnostic model. Model diagnostic efficacy was assessed through receiver operating characteristic (ROC) area under the curve (AUC) values and calibration curves. DeLong’s test was administered to validate model robustness.

Results

Our study encompassed 748 participants. Logistic regression modeling, trained with the aforementioned variables, demonstrated remarkable predictive capability, achieving an AUC of 0.805 (95% confidence interval (CI) [0.766–0.844]) in the primary cohort and 0.753 (95% CI [0.687–0.818]) in the validation cohort. Calibration plots suggested a favorable fit of the model to the data.

Conclusions

The developed logistic regression model emerges as a promising tool for forecasting EGFR mutation status. It holds potential to aid clinicians in more precisely identifying patients likely to benefit from EGFR molecular testing and facilitating targeted therapy decision-making, particularly in scenarios where molecular testing is impractical or inaccessible.

Introduction

Lung cancer is a common malignancy and a leading cause of cancer-related mortality worldwide (Siegel et al., 2023; Sung et al., 2021). Non-small cell lung cancer (NSCLC) is the most common histological subtype (Ahmad & Gadgeel, 2016; Ettinger et al., 2018). While surgical resection remains the primary treatment option for early-stage lung cancer, the insidious nature of the disease often leads to late diagnoses, rendering surgery impossible for many patients.

With advancements in research, targeted therapy has been developed for patients with positive driver gene mutations (Thai et al., 2021). This shift is driven by the improved progression-free survival (PFS) and quality of life observed in patients with epidermal growth factor receptor (EGFR) gene mutations who receive targeted biological therapies (Mitsudomi et al., 2010; Ramalingam et al., 2020; Tang et al., 2019).

In the latest studies, new therapeutic advancements have emerged, such as the introduction of targeted combinations like amivantamab and lazertinib (Cho et al., 2024). When used alongside chemotherapy, these combinations have demonstrated notable PFS benefits over chemotherapy alone. This progress highlights the expanding scope of personalized treatments that combine targeted and traditional approaches to further enhance patient outcomes.

Moreover, it is noteworthy that EGFR mutations are key drivers of NSCLC in Asian populations, making EGFR a critical target for genotype-driven therapy in this context (Melosky et al., 2021). This high prevalence further emphasizes the need for effective, targeted treatment options tailored to the specific genetic profiles of patients.

Currently, EGFR genetic testing relies on DNA sequencing of histological or cytological samples. However, obtaining such samples poses challenges, and tumor heterogeneity can compromise the accuracy of detection (Weber et al., 2014; Shahi et al., 2015; Zhang et al., 2019). While analyzing circulating tumor DNA (ctDNA) in plasma offers an alternative, its high false-negative rate and cost limit its widespread clinical application (Chang et al., 2018; Deng et al., 2020; Zhang et al., 2019). Consequently, the development of a non-invasive and highly accurate gene mutation detection method is crucial for clinical practice.

Recent studies have focused on developing predictive models that leverage clinical, radiological, and laboratory characteristics to ascertain EGFR mutation status in NSCLC.

Despite these advancements, practical application in clinical settings faces hurdles, including the limited patient populations in these studies and the extensive array of extracted features that may compromise generalizability (Jiang et al., 2019; Kim et al., 2024; Nair et al., 2020; Wang et al., 2022; Yang et al., 2022). Furthermore, many models rely on radiomics, requiring specialized software to extract computed tomography (CT) scan features, which is often inconsistent across studies and not widely adopted in routine clinical practice.

Epidemiological research consistently highlights correlations between EGFR mutations and specific patient profiles, including never-smokers, females, Asians, and those with advanced tumor stages (Girard et al., 2011; Russell et al., 2013; Shi et al., 2014). While numerous models integrating clinical and radiological features have emerged, some have ventured to incorporate serum tumor markers alongside radiological indicators to enhance EGFR mutation prediction. Nevertheless, the question persists: does a model that synthesizes all three categories of information—serum tumor markers, CT imaging traits, and clinical characteristics—offer superior diagnostic precision for EGFR mutation status? Surprisingly, there is a scarcity of models encompassing this holistic perspective.

To address this gap, our objective is to establish a more comprehensive and precise predictive model by collectively examining serum tumor markers, CT imaging features, and clinical attributes in NSCLC patients for forecasting EGFR gene mutations. Employing the least absolute shrinkage and selection operator (LASSO) method for variable selection ensures a rigorous and streamlined analytical approach. Participant data is derived from a large-scale, general hospital setting, thereby enhancing the potential real-world applicability and robustness of our model.

Materials and Methods

Study participants

This retrospective investigation encompasses patients diagnosed with NSCLC at the First Affiliated Hospital of Zhengzhou University between January 2018 and June 2019. The study was approved by the medical Ethics Committee of The First Affiliated Hospital of Zhengzhou University (approval no. 2023-KY-1100-002). Inclusion in the study mandated fulfillment of the following criteria: (1) a confirmed pathological diagnosis of NSCLC, (2) execution of CT scans within a month preceding surgical intervention or biopsy, (3) availability of full clinical records complemented by serum tumor marker assessments, and (4) an absence of prior malignancy histories.

Patients were excluded based on: (1) unavailability of EGFR genetic testing or test results, (2) prior receipt of anti-tumor therapies, and (3) incomplete clinical datasets, specifically lacking data on gender, age, smoking and alcohol consumption history, familial cancer history, past cancer occurrences, and tumor staging details. At our institution, EGFR genetic profiling was conducted via polymerase chain reaction (PCR) or next generation sequencing (NGS), employing histological or cytological specimens.

The compiled clinical dataset encompassed parameters such as gender, age, a familial background of cancer, personal cancer history, smoking and drinking habits, specific molecular pathology types, and thorough staging information of the tumors. Additionally, serum levels of select tumor markers were recorded, namely carcinoembryonic antigen (CEA) also in ng/ml, and cytokeratin 19 fragment (CYFRA21-1) in ng/ml. This comprehensive data assembly aimed to facilitate a meticulous evaluation of factors influencing EGFR mutation status prediction.

Image acquisition and feature extraction

The CT imaging characteristics were meticulously evaluated by a pair of senior radiologists employing a double-blind approach via the Picture Archiving and Communication System (PACS). In instances where disagreements arose, an additional, more seasoned physician was enlisted to contribute to the analysis, ultimately facilitating a consensus outcome. The parameters documented encompassed: nodule diameter (measured as the utmost transverse dimension in millimeters), presence or absence of emphysema, nodule positioning within the lung, total count of nodules identified, nodule morphology (classified as solid, part-solid, or purely ground-glass opacity), marginal characteristics (defined as irregular or regular), evidence of cavitation, observation of the vacuole sign, presence of spiculation, lobulation indication, pleural tail sign occurrence, calcification detection, assessment of vascular convergence patterns, identification of air bronchograms, differentiation between central and peripheral nodules, and detection of mediastinal lymphadenopathy.

Procedure of feature selection and predictive model

The bidirectional selection technique was employed to refine variables for the logistic regression model, guided by the Akaike Information Criterion (AIC) (Collins et al., 2015), ensuring a rigorous variable inclusion process. Turning to the establishment of the random forest model, the feature selection was entrusted to the advanced Boruta algorithm, known for its efficacy in attribute importance assessment.

In the case of the support vector machine (SVM) model, the kernel function type was meticulously screened utilizing the AUC metric. Subsequently, model accuracy served as the litmus test for determining the most favorable SVM configuration.

When constructing the eXtreme Gradient Boosting (XGBoost) model, a meticulous tuning of parameters such as Nrounds, max_depth, eta, gamma, colsample_bytree, min_child_weight, and subsample was conducted to optimize its performance, thereby extracting the model’s full potential.

Furthermore, both the SVM and XGBoost models benefited from an additional layer of variable curation through the application of LASSO, a robust method that further refined the predictive features, enhancing the overall precision and interpretability of these models. This rigorous, multipronged approach to feature selection and model optimization underscores a commitment to achieving the highest predictive accuracies possible.

Statistical analysis

All statistical analyses were conducted using R software version 4.2.3 (R Core Team, 2022). Continuous variables were expressed as mean ± standard deviation or as the median with interquartile ranges, depending on the observed data distribution. To evaluate differences between groups for continuous measures, either the independent t-test or the Mann–Whitney U test was employed, with the choice informed by the underlying data distribution. Discrete variables were analyzed for significant group discrepancies using the Chi-square test or Fisher’s exact test where appropriate.

A total of 30 cases had missing data, mainly due to the absence of initial testing for CEA and CYFRA21-1. Of these, four cases had both markers missing, one case had only CEA missing, and 25 cases had CYFRA21-1 missing. Multiple imputation methods were employed to address the missing data.

Incorporating serum tumor markers into our models necessitated their transformation via natural logarithms, adopting the formula: ln (serum tumor marker level) in units of ng/ml or U/ml. The predictive performance of the developed models was evaluated by calculating the area under the curve (AUC) from receiver operating characteristic (ROC) curves. To ensure model stability, DeLong’s test was applied. Additionally, we assessed the models’ performance using objective evaluation metrics, including specificity, precision, sensitivity, positive predictive value (PPV), negative predictive value (NPV), and F1-score. These metrics are derived from the models’ prediction outcomes, based on true positive (TP), true negative (TN), false negative (FN), and false positive (FP) values.

All statistical assessments were two-tailed, and a threshold of p < 0.05 was adopted to denote statistical significance throughout the analyses. This rigorous analytical framework aimed at yielding robust and reliable insights from the data.

Results

Clinical characteristics of patients

The study encompassed a total of 648 participants, who were stratified into a training cohort comprising 499 patients and a temporal external validation cohort consisting of 249 patients. This allocation was based on the date of biopsy or the surgical timeline. The training cohort represented 77% of the participants, while the validation cohort constituted the remaining 23%, ensuring a balanced and representative sample distribution (see Table 1).

Table 1 Clinical features and serum tumor markers of all patients.

	Primary cohort	p-value	Validation cohort	p-value	
	EGFR-wild type	EGFR-mutant		EGFR-wild type	EGFR-mutant		
Subjects (n)	n = 202	n = 297		n = 91	n = 158		
Sex:			<0.001			<0.001	
Male	138 (68.3%)	105 (35.4%)		68 (74.7%)	63 (39.9%)		
Female	64 (31.7%)	192 (64.6%)		23 (25.3%)	95 (60.1%)		
Age/years	64.0 [54.0;70.0]	61.0 [54.0;68.0]	0.023	65.0 [57.0;69.0]	60.0 [53.0;66.8]	0.021	
Smoking:			<0.001			<0.001	
Yes	102 (50.5%)	54 (18.2%)		52 (57.1%)	31 (19.6%)		
No	100 (49.5%)	243 (81.8%)		39 (42.9%)	127 (80.4%)		
Drinking:			<0.001			0.003	
Yes	59 (29.2%)	28 (9.43%)		29 (31.9%)	24 (15.2%)		
No	143 (70.8%)	269 (90.6%)		62 (68.1%)	134 (84.8%)		
Tumor history:			0.065			0.705	
Yes	19 (9.4%)	46 (15.5%)		12 (13.2%)	25 (15.8%)		
No	183 (90.6%)	251 (84.5%)		79 (86.8%)	133 (84.2%)		
Family history of cancer:			0.976			0.548	
Yes	35 (17.3%)	53 (17.8%)		17 (18.7%)	36 (22.8%)		
No	167 (82.7%)	244 (82.2%)		74 (81.3%)	122 (77.2%)		
Pathology type:			<0.001			<0.001	
Non-AC	60 (29.7%)	16 (5.39%)		29 (31.9%)	8 (5.06%)		
AC	142 (70.3%)	281 (94.6%)		62 (68.1%)	150 (94.9%)		
T stage:			<0.001			<0.001	
1	78 (38.6%)	156 (52.5%)		30 (33.0%)	94 (59.5%)		
2	71 (35.1%)	102 (34.3%)		33 (36.3%)	41 (25.9%)		
3	32 (15.8%)	29 (9.76%)		20 (22.0%)	20 (12.7%)		
4	21 (10.4%)	10 (3.37%)		8 (8.79%)	3 (1.90%)		
N stage:			0.256			0.273	
1	30 (19.9%)	58 (27.4%)		10 (15.4%)	25 (25.3%)		
2	46 (30.5%)	57 (26.9%)		25 (38.5%)	30 (30.3%)		
3	75 (49.7%)	97 (45.8%)		30 (46.2%)	44 (44.4%)		
M stage:			0.008			0.458	
1	100 (49.5%)	184 (62.0%)		42 (46.2%)	82 (51.9%)		
0	102 (50.5%)	113 (38.0%)		49 (53.8%)	76 (48.1%)		
TNM stage:			0.003			0.063	
I	38 (18.8%)	57 (19.2%)		18 (19.8%)	45 (28.5%)		
II	14 (6.93%)	19 (6.40%)		7 (7.69%)	9 (5.70%)		
III	50 (24.8%)	37 (12.5%)		24 (26.4%)	22 (13.9%)		
IV	100 (49.5%)	184 (62.0%)		42 (46.2%)	82 (51.9%)		
CEA (ng/ml)	4.02 [2.02;17.9]	7.56 [2.93;44.3]	0.003	4.51 [2.58;14.2]	4.22 [2.26;18.4]	0.988	
CYFRA 21-1 (ng/ml)	4.25 [2.45;9.14]	2.83 [1.78;5.75]	<0.001	3.63 [2.39;6.41]	3.08 [2.04;4.99]	0.158	
Note:

Data are presented as median (with interquartile range) n (%). Non-AC, non-adenocarcinoma; AC, adenocarcinoma; CEA, carcinoembryonic antigen; CYFRA 21-1, cytokeratin 19 fragment.

Notably, EGFR mutation-positive cases predominantly comprised females who demonstrated significantly less exposure to both smoking and alcohol compared to individuals in the EGFR wild-type group, highlighting distinct demographic and lifestyle correlations (as detailed in Table 1).

Table 2 provides an overview of the prevalent imaging characteristics observed across the cohorts. Furthermore, as illustrated in Fig. 1, a substantial 80.6% of the enrolled patients were found to harbor classical EGFR mutations, specifically L858R mutations and/or exon 19 deletions (Ex19del), with some cases presenting concurrently with T790M mutations, underscoring the predominance of these genetic alterations in the studied population.

Table 2 General imaging features of all patients.

	Primary cohort	p-value	Validation cohort	p-value	
	EGFR-wild type	EGFR-mutant		EGFR-wild type	EGFR-mutant		
Subjects (n)	n = 202	n = 297		n = 91	n = 158		
Emphysema:			<0.001			<0.001	
Yes	73 (36.1%)	40 (13.5%)		41 (45.1%)	8 (5.06%)		
No	129 (63.9%)	257 (86.5%)		50 (54.9%)	150 (94.9%)		
Diameter (mm)	36.0 [24.0;51.5]	28.0 [20.0;39.0]	<0.001	38.0 [27.0;57.5]	27.0 [20.0;38.0]	<0.001	
Nodule location:			0.310			0.758	
Left upper lobe	56 (27.7%)	70 (23.6%)		24 (26.4%)	42 (26.6%)		
Left lower lobe	30 (14.9%)	39 (13.1%)		12 (13.2%)	26 (16.5%)		
Right upper lobe	52 (25.7%)	102 (34.3%)		29 (31.9%)	46 (29.1%)		
Right middle lobe	15 (7.43%)	16 (5.39%)		5 (5.49%)	14 (8.86%)		
Right lower lobe	49 (24.3%)	70 (23.6%)		21 (23.1%)	30 (19.0%)		
Number:			0.513			0.300	
1	115 (56.9%)	159 (53.5%)		68 (74.7%)	103 (65.2%)		
2	29 (14.4%)	38 (12.8%)		7 (7.69%)	12 (7.59%)		
3	5 (2.48%)	5 (1.68%)		0 (0.00%)	3 (1.90%)		
More than 3	53 (26.2%)	95 (32.0%)		16 (17.6%)	40 (25.3%)		
Type:			0.082			0.324	
Solid	195 (96.5%)	272 (91.6%)		87 (95.6%)	142 (89.9%)		
Pure ground-glass	5 (2.48%)	20 (6.73%)		2 (2.20%)	10 (6.33%)		
Part-solid	2 (0.99%)	5 (1.68%)		2 (2.20%)	6 (3.80%)		
Margin:			0.604			0.625	
Irregular	194 (96.0%)	281 (94.6%)		89 (97.8%)	156 (98.7%)		
Regular	8 (3.96%)	16 (5.39%)		2 (2.20%)	2 (1.27%)		
Cavitation:			0.745			0.564	
Yes	42 (20.8%)	57 (19.2%)		9 (9.89%)	11 (6.96%)		
No	160 (79.2%)	240 (80.8%)		82 (90.1%)	147 (93.0%)		
Vocule sign:			0.957			0.926	
Yes	24 (11.9%)	37 (12.5%)		8 (8.79%)	12 (7.59%)		
No	178 (88.1%)	260 (87.5%)		83 (91.2%)	146 (92.4%)		
Burr sign:			0.386			0.819	
Yes	131 (64.9%)	180 (60.6%)		39 (42.9%)	64 (40.5%)		
No	71 (35.1%)	117 (39.4%)		52 (57.1%)	94 (59.5%)		
Lobulation:			0.283			1.000	
Yes	157 (77.7%)	217 (73.1%)		59 (64.8%)	102 (64.6%)		
No	45 (22.3%)	80 (26.9%)		32 (35.2%)	56 (35.4%)		
Pleural traction			0.280			<0.001	
Yes	125 (61.9%)	199 (67.0%)		41 (45.1%)	110 (69.6%)		
No	77 (38.1%)	98 (33.0%)		50 (54.9%)	48 (30.4%)		
Calcification:			0.366			0.039	
Yes	15 (7.43%)	15 (5.05%)		17 (18.7%)	14 (8.86%)		
No	187 (92.6%)	282 (94.9%)		74 (81.3%)	144 (91.1%)		
Vascular convergence:			0.730			0.002	
Yes	183 (90.6%)	265 (89.2%)		82 (90.1%)	156 (98.7%)		
No	19 (9.41%)	32 (10.8%)		9 (9.89%)	2 (1.27%)		
Air bronchogram:			0.630			0.474	
Yes	57 (28.2%)	91 (30.6%)		41 (45.1%)	80 (50.6%)		
No	145 (71.8%)	206 (69.4%)		50 (54.9%)	78 (49.4%)		
Central or Peripheral:			0.269			0.040	
Central	73 (36.1%)	92 (31.0%)		7 (7.69%)	3 (1.90%)		
Peripheral	129 (63.9%)	205 (69.0%)		84 (92.3%)	155 (98.1%)		
Mediastinal lymphadenopathy:			0.625			0.005	
Yes	106 (52.5%)	148 (49.8%)		49 (53.8%)	55 (34.8%)		
No	96 (47.5%)	149 (50.2%)		42 (46.2%)	103 (65.2%)		
Note:

Data are presented as median (with interquartile range) n (%).

Figure 1 The proportion of EGFR classical and non-classical mutations in enrolled NSCLC patients.

Classical EGFR mutations are L858R and various Ex19dels.

A logistic regression model was derived, optimized for the minimum AIC, incorporating nine predictive variables. Meanwhile, the random forest model was constructed with a total of eleven variables—nine deemed definitively important and two provisionally so—based on their Boruta algorithm-derived importance values. The random forest model’s parameter mtry was fixed at one to achieve peak accuracy.

LASSO regression, following a 10-fold cross-validation with the 1-SE rule, led to the identification of eight risk predictors using a lambda value of 0.046. These selected variables were subsequently employed in both the SVM and XGBoost models. The SVM model, utilizing a radial basis function kernel, was fine-tuned with sigma = 0.1 and cost = 0.1 for optimal performance.

As for the XGBoost model, a grid search determined the optimal parameters to be: nrounds = 100, max_depth = 3, eta = 0.1, gamma = 0.1, colsample_bytree = 0.8, min_child_weight = 3, and subsample = 0.8.

Model predictive performances were assessed via metrics including the AUC, accuracy, sensitivity, and specificity (summarized in Table 3). Notably, the XGBoost model excelled with an AUC of 0.907 (95% CI [0.880–0.933]) in the primary cohort, albeit showing reduced performance in the validation cohort with an AUC of 0.764 (95% CI [0.698–0.829]). The Delong test revealed significant instability with a p-value of 8.71e−05, echoed by similar instability noted in the random forest model.

Table 3 The predictive performance of various models in the primary and validation.

	AUC (95% CI)	Accuracy	Sensitivity	Specificity	p for Delong	
Logistical regression					0.18	
Primary	0.805 [0.766–0.844]	0.762 (0.721–0.798)	0.765	0.755		
Validation	0.753 [0.687–0.818]	0.711 (0.650–0.766)	0.756	0.617		
Random forest					0.002	
Primary	0.882 [0.852–0.912]	0.802 (0.764–0.836)	0.939	0.599		
Validation	0.772 [0.708–0.836]	0.751 (0.693–0.803)	0.861	0.560		
Support vector machine					0.69	
Primary	0.741 [0.703–0.779]	0.776 (0.736, 0.811)	0.923	0.559		
Validation	0.727 [0.670–0.785]	0.763 (0.705, 0.815)	0.861	0.593		
Xgboost					<0.001	
Primary	0.907 [0.880–0.933]	0.832 (0.796, 0.863)	0.919	0.703		
Validation	0.764 [0.698–0.829}	0.743 (0.684, 0.7961)	0.785	0.670		
Note:

AUC, area under the receiver operating characteristic curve; CI, confidence interval.

Consequently, considering stability and overall performance, the logistic regression model was deemed the most suitable. Comparative ROC curves for all models are depicted in Fig. 2.

Figure 2 The ROC curves of various models.

(A) Receiver operating characteristic curves of various models in the primary cohort. (B) Receiver operating characteristic curves of various models in the validation cohort.

Optimal model analysis

Regression equation of our model:

X = 0.239 – 0.019 × diameter (mm) + (0.468 × TNM stage I/−0.058 × TNM stage III/0.855 × TNM stage IV) + 0.525 × tumor history – 1.042 × smoking history – 0.642 × emphysema − (0.501 × right lower lobe/1.221 × right middle lobe/0.600 × left upper lobe/0.246 × left lower lobe) + 1.533 × adenocarcinoma + 0.196 × Ln (CEA) – 0.441 × Ln (CYFRA 21-1), P=exp(X)/(1+exp(X)).

For clarification, in the TNM staging, Stage II is represented by a value of 0, while Stages I, III, and IV equate to 1. Regarding nodule location, the right upper lobe is encoded as 0, and all other locations as 1. Categorical variables are assigned a value of 1 if the condition is present and 0 if absent.

This multivariate logistic regression model is visually presented as a forest plot in Fig. 3. Additionally, the relative importance of the selected variables associated with EGFR mutation status was assessed using an XGBoost machine learning model, and the findings are illustrated in Fig. 4. A nomogram (Fig. 5) was constructed based on the logistic regression model for predicting EGFR mutation probabilities. An illustrative example is a 73-year-old male validation cohort patient with stage IV non-adenocarcinoma, a history of smoking, no prior tumor history, and a nodule in the right upper lobe accompanied by emphysema. His probability of harboring an EGFR mutation, depicted by a red dot in the relevant figure, underscores the model’s predictive capability.

Figure 3 Forest diagram of the logistic regression model.

Figure 4 XGBoost model elucidates the relative significance of each variable which included in the logistic regression model on the mutation status of EGFR and the corresponding variable significance score.

The X-axis indicates the importance score which is the relative number of a variable that is used to distribute the data; the Y-axis is the selected variables.

Figure 5 Nomogram to estimate the odds of EGFR mutation in NSCLC patients.

The red dot represents each value of the included variable of one patient in the validation cohort, and the red interval indicating 95% confidence interval of the probability of EGFR mutation for this patient. RUL, right upper lobe. RML, right middle lobe. RLL, right lower lobe. LUL, left upper lobe. LLL, left lower lobe. ***p < 0.001; **p < 0.01; *p < 0.05.

The predictive performance of the logistic regression model is detailed in Table 4. The model’s calibration in both the primary and validation cohorts was evaluated using a calibration curve with 1,000 bootstrapped resamples, as shown in Fig. 6. These curves confirm that the predicted probabilities of EGFR mutations closely match the actual observations in both cohorts.

Table 4 Predictive performance of the logistic regression model.

	Sensitivity (95% CI)	Specificity (95% CI)	PPV (95% CI)	NPV (95% CI)	F1-score (95% CI)	
Primary	0.765 [0.716–0.809]	0.755 [0.681–0.819]	0.865 [0.821–0.902]	0.609 [0.538–0.677]	0.812 [0.778–0.849]	
Validation	0.756 [0.684–0.819]	0.617 [0.503–0.723]	0.804 [0.733–0.863]	0.549 [0.442–0.654]	0.779 [0.830–0.734]	
Note:

PPV, positive predictive value; NPV, negative predictive value; CI, confidence interval.

Figure 6 The calibration curves of our prediction model in the primary and validation cohort.

(A) The calibration curve of the primary cohort. (B) The calibration curve of the validation cohort.

Lastly, Fig. 7 presents the decision curves for the logistic regression model in the primary and validation cohorts. The curves demonstrate that when the threshold probability for EGFR mutation is set at 10%, applying our logistic regression model to predict EGFR mutation status in NSCLC patients yields more net benefit compared to strategies of either universal treatment or no treatment at all (Huang et al., 2016), particularly evident with a wider net benefit range observed in the validation cohort, highlighting the clinical utility and practical applicability of our predictive model.

Figure 7 Decision curve analysis for the logistic regression model in the primary and validation cohort.

Discussion

With targeted therapies becoming a cornerstone in the oncological management of NSCLC (Imyanitov, Iyevleva & Levchenko, 2021), the detection of EGFR gene mutations has garnered significant interest. This is primarily due to the high prevalence of such mutations among NSCLC patients and their crucial role in predicting therapeutic responses to EGFR-specific tyrosine kinase inhibitors (TKIs) (Dang et al., 2020; Giulio et al., 2006; Lindeman et al., 2018). However, the reliance on invasive biopsy procedures for EGFR mutation testing poses challenges, preventing some patients from undergoing this crucial diagnostic step in clinical practice. This gap underscores the urgent requirement for a non-invasive approach to predict EGFR mutations in NSCLC.

In recent times, numerous predictive models leveraging radiomics have emerged; however, their widespread adoption has been hindered by the software dependencies involved in feature extraction, limiting international accessibility. Our study addresses this issue by retrospectively assembling a dataset comprising clinical parameters, serum tumor marker levels, and quantifiable CT imaging attributes from NSCLC patients. Through meticulous analysis of this compiled information, we have devised an innovative predictive model tailored to anticipate EGFR mutation status without necessitating invasive procedures. This endeavor aims to facilitate more inclusive and less intrusive mutation testing, thereby enhancing personalized treatment strategies in NSCLC management.

In this study, we developed and evaluated four machine learning models to predict EGFR mutation status in NSCLC patients using readily available clinical data, serum tumor markers, and CT imaging features. Model performance was rigorously assessed based on AUC, accuracy, sensitivity, specificity, and stability using the DeLong test.

The conventional logistic regression model emerged as the most robust and stable predictor, demonstrating excellent calibration. This was evidenced by the strong agreement between predicted and actual probabilities, confirmed by non-significant Hosmer-Lemeshow test results (p > 0.05) in both the training and validation cohorts. Decision curve analysis and a nomogram further highlighted the model’s clinical utility and net benefit.

In Fig. 4, the relative importance of each variable within the logistic regression model for predicting EGFR mutation status is depicted, as inferred from the XGBoost algorithm. Among these, the top five influential factors identified were the levels of CYFRA 21-1 and CEA, nodule diameter, smoking status, and adenocarcinoma diagnosis. Our findings align with Zhang et al.’s (2022) research, which also observed elevated CYFRA 21-1 levels in EGFR wild-type patients.

Consistent with past literature suggesting CEA as a potential biomarker for adenocarcinoma due to its heightened expression in such cases (Jin et al., 2014; Molina et al., 1994; Yoshino et al., 2006), CEA emerged as another significant predictor in our model. We further corroborated earlier studies revealing a positive correlation between smaller lesion diameters and increased likelihood of EGFR mutations (Hsu et al., 2014; Rizzo et al., 2015).

Previous research has consistently linked EGFR mutations with female gender, non-smoking status, and adenocarcinoma histology (Chang et al., 2018; Chapman et al., 2016; Duma, Santana-Davila & Molina, 2019; Jemal et al., 2018; Kawaguchi et al., 2016; Shi et al., 2014); however, gender was not retained in our final logistic regression model. Despite this, our cohorts reflected a higher prevalence of EGFR mutations among females, approximately 60%, echoing the established gender disparity. This contrasts with the findings of Chen et al. (2019). One plausible explanation for gender’s exclusion could be the inclusion of adenocarcinoma as a predictor, as other histological types in our sample less frequently underwent EGFR testing.

Regarding nodule location, our data highlighted the right upper lobe as an independent predictor of EGFR mutation. Byers, Vena & Rzepka (1984) among others, has noted a predilection for lung cancers to occur in upper lobes, partially explaining this observation, though further investigation is merited to elucidate the precise mechanisms and associations.

Advanced disease stage, previously implicated as a pivotal predictor of EGFR mutation in NSCLC (Girard et al., 2011), was similarly identified as an independent predictor in our study. Emphysema’s inclusion might stem from the recognized association between adenocarcinoma, which has a higher incidence of EGFR mutations, and emphysema, compared to squamous cell carcinoma. Overall, our model underscores the multifactorial nature of EGFR mutation prediction and invites deeper exploration into the underlying biological and clinical correlations.

The primary dataset demonstrated superior overall performance compared to the validation dataset, achieving higher values for specificity, PPV, NPV, and F1-score. Specifically, the primary dataset achieved an F1-score of 0.812, while the validation dataset recorded an F1-score of 0.779. Notably, the PPV in the primary dataset (0.865) was significantly higher than that in the validation set (0.804). However, sensitivity remained fairly consistent across both datasets, with values around 0.76.

These findings indicate that while the model performs reasonably well in both datasets, there is some degradation in the validation set, particularly in terms of specificity and NPV. This suggests opportunities for refinement when the model is applied to new data.

Our research integrated clinical characteristics, CT imaging traits, and tumor marker levels into a predictive framework. Distinguishing itself from alternative predictive models, our study leveraged easily accessible data from routine clinical practices, eliminating the need for supplementary experiments or specialized software to extract CT characteristics. The robustness of our model’s discriminative ability in forecasting EGFR mutations was validated through an external validation cohort, thereby reinforcing its potential utility in guiding targeted therapies, especially in scenarios where comprehensive molecular profiling is constrained, and aiding clinicians in deciding upon the appropriateness of molecular testing.

Despite these strengths, several limitations warrant acknowledgment. Firstly, while an internal validation cohort was constituted, its sample size was modest, potentially limiting the generalizability of our findings. Secondly, the retrospective nature of our study design introduces a possibility of selection bias among the included patients, necessitating cautious interpretation of the results. To mitigate these biases and enhance the model’s validity, future research should encompass large-scale, multicenter, and prospective studies. Lastly, our model did not incorporate randomization procedures, and future iterations could benefit from exploring a broader array of predictive factors to further enhance the discriminatory power and predictive accuracy. These enhancements will fortify the model’s clinical utility and reliability in guiding personalized treatment strategies for NSCLC patients.

Conclusions

In this study, we have successfully developed and validated a predictive model for determining EGFR mutation status in NSCLC patients. This model, based on logistic regression and integrating clinical parameters, CT characteristics, and serum tumor markers, demonstrated significant predictive power with AUC values of 0.805 in the primary cohort and 0.753 in the validation cohort. The model’s high discriminatory ability and accurate calibration highlight its robustness and reliability.

The practical application of this model is particularly promising for clinical settings where molecular testing is not feasible due to resource constraints or accessibility issues. By identifying patients who are likely to benefit from EGFR molecular testing, our model supports more targeted and effective decision-making for EGFR-directed therapies. This enables a more personalized approach to treatment, optimizing the use of available resources and potentially improving patient outcomes.

In summary, our predictive model offers a valuable tool for enhancing EGFR mutation status forecasting. It not only contributes to more efficient clinical decision-making but also supports a tailored approach to NSCLC management, ultimately benefiting patient care and resource allocation.

Supplemental Information

Supplemental Information 1 Raw data.

The authors would like to thank all the participants who took part in this study. We also extend our gratitude to Liwei Zhou for securing ethics approval and for his involvement in data collection.

Additional Information and Declarations

Competing Interests

Author Contributions

Human Ethics

Data Availability

The authors declare that they have no competing interests.

Jimin Hao conceived and designed the experiments, performed the experiments, analyzed the data, prepared figures and/or tables, authored or reviewed drafts of the article, and approved the final draft.

Man Liu performed the experiments, prepared figures and/or tables, authored or reviewed drafts of the article, and approved the final draft.

Zhigang Zhou conceived and designed the experiments, performed the experiments, authored or reviewed drafts of the article, and approved the final draft.

Chunling Zhao performed the experiments, prepared figures and/or tables, and approved the final draft.

Liping Dai conceived and designed the experiments, performed the experiments, analyzed the data, authored or reviewed drafts of the article, and approved the final draft.

Songyun Ouyang conceived and designed the experiments, performed the experiments, analyzed the data, authored or reviewed drafts of the article, and approved the final draft.

The following information was supplied relating to ethical approvals (i.e., approving body and any reference numbers):

The medical Ethics Committee of the first affiliated hospital of Zhengzhou University.

The following information was supplied regarding data availability:

The raw measurements are available in the File S1.

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
