# Peer review of "Predicting epidermal growth factor receptor (EGFR) mutation status in non-small cell lung cancer (NSCLC) patients through logistic regression: a model incorporating clinical characteristics, computed tomography (CT) imaging features, and tumor marker levels"

_PeerJ, doi:10.7717/peerj.18618_

## Round 0.1 · original submission · Major Revisions

The authors are requested to carefully revise the manuscript and answer the questions raised by the reviewers.

·

Basic reporting

None

Experimental design

None

Validity of the findings

None

Additional comments

This article focuses on the development and validation of a logistic regression model using clinical features, CT imaging features, and tumor marker levels to predict EGFR mutation status in patients with NSCLC. We highly appreciate the authors' work; the experimental design is generally reasonable, the research ideas are clear, and the logic chain is complete. However, after careful review, we would like to decide to reject the manuscript in its current form due to several significant methodological and reporting issues outlined below.
1. Patient selection criteria are unclear and potentially biased. More details are needed on the inclusion/exclusion criteria, particularly around selection of patients for EGFR testing. To avoid bias, the study should aim to include all NSCLC patients, not just those who underwent testing. The reasons for not testing should be reported.
2. Handling of missing data is not described. The amount of missing data and the statistical methods used to account for this should be clearly reported, as it can significantly impact the validity of the results.
3. With a large number of candidate predictors and the use of stepwise variable selection, the risk of overfitting the models to the dataset is high. More rigorous methods to limit overfitting should be employed.
4. The authors' affiliations are inconsistent with the department that granted ethical approval. All authors are from the Department of Respiratory and Sleep Medicine, Department of Radiology, or Institute of Tumor Molecular Markers, and none of them are affiliated with the Nutrition Department. This raises doubts about whether the study truly followed appropriate ethical review and approval procedures. The authors need to clarify the ethical review process and provide documentation of proper approval by a qualified ethics committee.

·

Basic reporting

The authors created a prediction model for EGFR mutations in NSCLC based on clinical, CT, and serum marker data. Their logistic regression model performed well in both the training and validation populations, indicating that it might be a useful tool for guiding treatment decisions.

Experimental design

See below

Validity of the findings

Firstly, Although the proposed study is successful in terms of organization, presentation, content and results, major revision given in the following items need to be performed.
1) Improve the conclusion section, enhance the manuscript to convey the purpose, objectives, method and major findings.
2) Use abbreviations after the first use in the text, in the abstract and throughout the paper.
3) The literature review is quite insufficient in the introduction section. Complete the introduction and literature sections of the article by providing similar studies from the years 2023-2024 and/or new and current studies that will attract the attention of the readers.
4) Compare Logistic regression based machine learning methods with more performance metrics than ROC and AUC. More analysis results should be included in the results and findings section.
5) The resolution of the figures giving the analysis results should be increased.
6) Clean the paper of English spelling and punctuation errors

---

## Round 0.2 · accepted · Accept

After revisions, all reviewers agreed to publish the manuscript. I also reviewed the manuscript and found no obvious risks to publication. Therefore, I also approved the publication of this manuscript.

·

Basic reporting

After reviewing the authors' responses to the reviewer comments and the revised manuscript, I believe the authors have adequately addressed all the major concerns raised. They have clarified the patient selection criteria, provided an appropriate method for handling missing data, and employed rigorous strategies to prevent model overfitting. Additionally, the ethical approval issue has been resolved, with the ethics application process clarified and the responsible individual added as a co-author.
In conclusion, the authors have adequately addressed the reviewer's suggestions and resolved the major issues that were previously raised. Therefore, I believe the manuscript is now suitable for publication.

Experimental design

--

Validity of the findings

--

Additional comments

After reviewing the authors' responses to the reviewer comments and the revised manuscript, I believe the authors have adequately addressed all the major concerns raised. They have clarified the patient selection criteria, provided an appropriate method for handling missing data, and employed rigorous strategies to prevent model overfitting. Additionally, the ethical approval issue has been resolved, with the ethics application process clarified and the responsible individual added as a co-author.
In conclusion, the authors have adequately addressed the reviewer's suggestions and resolved the major issues that were previously raised. Therefore, I believe the manuscript is now suitable for publication.